# Impact of water uptake on fluorescence of atmospheric aerosols: Insights from Mie-Raman-Fluorescence lidar measurements

Igor Veselovskii<sup>1</sup>, Qiaoyun Hu<sup>2</sup>, Philippe Goloub<sup>2</sup>, Thierry Podvin<sup>2</sup>, Gaël Dubois<sup>2</sup>, Alexey Kolgotin<sup>1</sup>, Mikhail Korenskii<sup>1</sup>

<sup>1</sup>Prokhorov General Physics Institute of the Russian Academy of Sciences, Moscow, Russia <sup>2</sup>Univ. Lille, CNRS, UMR 8518 - LOA - Laboratoire d'Optique Atmosphérique, F-59650 Lille, France Correspondence to: Igor Veselovskii (igorv@pic.troitsk.ru)

Abstract. This study investigates the impact of water uptake by particles during hygroscopic growth on aerosol fluorescence properties, using multi-wavelength Mie-Raman-Fluorescence lidar measurements conducted at the ATOLL observatory (Laboratoire d'Optique Atmosphérique, University of Lille) between 2021 and 2024. During certain episodes we observed a systematic height-dependent decrease in the fluorescence backscattering coefficient within the well-mixed planetary boundary layer. This phenomenon begins at relatively low relative humidity (RH  $\sim$ 50%) simultaneously with decreasing the particle depolarization ratio. However, the rapid growth of aerosol backscattering coefficient at high RH is not mirrored by the same rate of fluorescence reduction. This distinct behaviour suggests a nonlinear relationship between water uptake and fluorescence suppression, likely indicating that water-induced quenching effects operate independently of bulk hygroscopic growth. Furthermore, we demonstrate the capability to retrieve particle volume and surface area density from single-wavelength extinction coefficients during strong hygroscopic growth episodes, validated against full  $3\beta$ +2 $\alpha$  lidar measurements. The values of the conversion factors for urban aerosol and smoke at 355 nm and 532 nm, together with associated uncertainties, are presented.

#### 20 1 Introduction

Atmospheric aerosols are the subject of intense study due to their role in the planet's radiation balance and their potential impact on the environment. To characterize the vertical distribution of aerosol loading, multiwavelength Mie-Raman and high spectral resolution lidars are widely used (e.g., Burton et al., 2012; Floutsi et al., 2023, Haarig et al., 2025 and references therein). Studies conducted over the past two decades have demonstrated significant progress in retrieving particle microphysical properties such as volume density, effective radius, and complex refractive index (RI) from so-called  $3\beta+2\alpha$  observations, which include three aerosol backscattering,  $\beta$ , and two extinction,  $\alpha$  coefficients (Veselovskii et al., 2002; Kolgotin et al., 2018, 2023, 2024; Chang et al., 2022; Zhou et al., 2024).

The  $3\beta+2\alpha$  dataset is not always available. However, if the aerosol type is identified, the particle volume and surface area densitiy can be estimated from the aerosol extinction coefficient at a single wavelength,  $\lambda$ , using corresponding extinction-to-volume and extinction-to-surface area conversion factors. This approach was proposed by Mamouri and Ansmann (2016,

2017) and further developed by Ansmann et al. (2021), where optical depths and column-integrated volumes provided by the network of sun photometers (AERONET) were used to determine conversion factors for pollution, smoke, and dust at various locations. Similarly, He et al. (2023) reported conversion factors for dust of different origins. However, the column-integrated values provided by the AERONET do not account for the possible presence of different aerosol types at different altitudes. In contrast, multiwavelength lidar measurements allow for the retrieval of vertical profiles of particle volume and surface area density, enabling the calculation of conversion factors within specific height intervals that correspond to distinct aerosol types. An important factor influencing the physical properties of aerosols is the relative humidity (RH). Many aerosol types, such as sulphates, organic carbon, and sea salt, increase in size at high RH, accompanied by a simultaneous decrease in their complex refractive index. Lidars, with their capability to assess particle properties under ambient conditions at RH levels close to saturation, offer unique opportunities for studying the particle hygroscopic growth. Lidar measurements are typically used to determine the dependence of the aerosol backscattering coefficient on RH within well-mixed layers (Feingold and Morley, 2003; Fernández et al., 2015; Granados-Muñoz et al., 2015; Haarig et al., 2017; Navas-Guzmán et al., 2019; Chen et al., 2019; Pérez-Ramírez et al., 2021; Sicard et al., 2022, Miri et al., 2024). The use of the aerosol extinction coefficients in hygroscopic growth studies is less common (Veselovskii et al., 2009; Dawson et al., 2020; Haarig et al., 2025), because  $\beta$  can be retrieved from lower altitudes and with greater accuracy compared to  $\alpha$ . Multiwavelength lidar measurements further extend hygroscopic growth studies by enabling the analysis of particle volume, surface area and the refractive index as a function of RH. This, in turn, allows for the evaluation of extinction-to-volume and extinction-to-surface area conversion factors in the presence of hygroscopic growth.

40

55

60

The addition of a fluorescence channel to Mie-Raman lidar (Veselovskii et al., 2020) opens new opportunities for particle characterization. If hygroscopic growth has no impact on aerosol fluorescence properties, fluorescence may serve as a proxy for evaluating the volume of dry aerosol. However, laboratory studies demonstrate that fluorescence can be suppressed ("quenched") by interactions with other molecules through processes of collision and energy transfer (Lakowicz, 2006), leading to reduction in emission intensity. In particular, water molecules can efficiently quench fluorescence of organic fluorophores as reported by Dobretsov et al. (2014) and Maillard et al. (2021). Meanwhile, the extent to which such quenching occurs in atmospheric aerosols during hygroscopic growth remains an open question (Gast et al., 2024; Reichardt et al., 2025). Fluorescence quenching is expected to depend on aerosol composition, phase state, and presence of organic coating, highlighting the need to analyse a diverse set of measurement episodes.

The Mie-Raman-Fluorescence lidar has been operational at the ATOLL (ATmospheric Observation at LiLLe) instrumentation site at the Laboratoire d'Optique Atmosphérique, University of Lille, since 2020, accumulating a large volume of observations. In this paper, we focus on several episodes of hygroscopic growth within the planetary boundary layer (PBL), investigating potential evidence of fluorescence quenching by water uptake. Additionally, we analyze extinction-to-volume and extinction-to-surface area conversion factors in the presence of hygroscopic growth. We begin with description of the lidar system and data analysis approach (Section 2). In Section 3 a numerical simulation is performed to investigate the behaviour of particle volume and surface area density as a function of the particle extinction coefficient for different refractive indices and different

particle size distribution parameters. In the first results section (Sections 4), we present case studies demonstrating that, in certain episodes, water uptake leads to fluorescence quenching. In Section 5, we analyse the conversion factors in the presence of the particle hygroscopic growth. The paper concludes with a summary of our findings in the conclusion section.

## 2 Instrumentation and methodology

The multiwavelength Mie-Raman-Fluorescence lidar LILAS (LIlle Lidar AtmosphereS) operates using a tripled Nd:YAG laser with a 20 Hz repetition rate and a pulse energy of 90 mJ at 355 nm. Backscattered light is collected by a 400 mm aperture Newtonian telescope. Detailed information on the system parameters and data analysis methodology can be found in our previous publications (e.g. Veselovskii et al., 2020). Measurements were performed primarily in the vertical direction. In cases where measurements were taken at an angle, this information is provided in the figure captions. LILAS is designed to detect elastic and Raman backscattering, enabling the so called  $3\beta+2\alpha+3\delta$  data configuration. This includes three particle backscattering ( $\beta_{355}$ ,  $\beta_{532}$ ,  $\beta_{1064}$ ), two extinction ( $\alpha_{355}$ ,  $\alpha_{532}$ ) coefficients and three particle linear depolarization ratios ( $\delta_{355}$ ,  $\delta_{532}$ ,  $\delta_{1064}$ ). Additionally, the water vapor mixing ratio (WVMR) is derived from the water vapor Raman measurements. The lidar also profiles the laser-induced fluorescence of aerosol particles within a spectral interval centered at 466 nm with 44 nm bandwidth. The fluorescence backscattering coefficient,  $\beta_F$ , and the fluorescence capacity,  $G_F = \frac{\beta_F}{\beta_{532}}$ , calculated from these

measurements, facilitate aerosol classification, as described in Veselovskii et al. (2024).

The inversion of  $3\beta+2\alpha$  lidar dataset enables the estimation of particle volume, V, surface area, S, and the refractive index (RI). In our study, we used the inversion algorithm described by Veselovskii et al. (2002). The solution search was performed within the following parameter ranges:

• Particle radius: 0.05<r<10 μm

80

85

• Real part of the refractive index:  $1.33 < m_R < 1.65$ 

• Imaginary part of the refractive index:  $0 < m_I < 0.02$ 

The retrieval uncertainty for particle volume and surface density is estimated to be below 15%, while for the real part of the refractive index it is  $\pm 0.05$ . The uncertainty for the imaginary part is higher, reaching up to 50%. Therefore, in the analysis of the presented measurements, only the real part of the refractive index is considered. Retrieved values of V and S along with  $3\beta+2\alpha$  set enable the calculation of conversion factors for both aerosol extinction and backscattering coefficients, defined as

$$C_{\alpha,\lambda}^{V} = \frac{V}{\alpha_{\lambda}}, C_{\beta,\lambda}^{V} = \frac{V}{\beta_{\lambda}}, C_{\alpha,\lambda}^{S} = \frac{S}{\alpha_{\lambda}}$$
 and  $C_{\beta,\lambda}^{S} = \frac{S}{\beta_{\lambda}}$ . These factors allow for the estimation of  $V$  and  $S$  from a single extinction

coefficient at wavelength  $\lambda$ , when the aerosol type is known.

To assess the effect of water uptake on fluorescence, we selected episodes where the PBL was well mixed, ensuring that aerosol composition and particle number density remained unchanged with height. A well-mixed layer was identified based on stable WVMR and potential temperature. The potential temperature was available from radiosonde measurements at

Herstmonceux (UK) and Beauvechain (Belgium) stations, located 160 km and 80 km away from the observation site, respectively. The relative humidity profiles were calculated using lidar-derived WVMR and temperature profiles from radiosonde data and from Global Data Assimilation System (GDAS). Across all episodes analysed in this work, the difference between RH profiles derived from radiosonde temperatures and temperatures from GDAS remained below 10%.

Within the well-mixed PBL, relative humidity increases with height, leading to an increase in the aerosol extinction and backscattering coefficients. However, these coefficients respond differently to changes in RH. Water uptake reduces the particle refractive index, but the extinction coefficient is less sensitive to the refractive index changes than backscattering. As a result, during hygroscopic growth  $\alpha$  increases more rapidly with RH than  $\beta$ , causing the extinction-to-backscattering ratio (lidar ratio) to rise (Haarig et al., 2025).

Dependence of backscattering on RH can be either monotonic or deliquescent, when backscattering is unchanged up to a certain RH value, called the deliquescence point, where a phase transition occurs from solid to liquid (Carrico et al., 2003; Randriamiarisoa et al., 2006). For relative humidity exceeding the deliquescence value, the dependence of particle parameters, such as  $\beta$ ,  $m_R$ , and r on RH can be modeled on relative to their values at a some reference height,  $z_{ref}$ . In particular, for the backscattering coefficient the simplified Hänel parameterization is commonly used (Hänel, 1976; Sheridan et al., 2002; Sicard et al., 2022):

110 
$$\frac{\beta}{\beta_{ref}} = \left(\frac{1 - RH}{1 - RH_{ref}}\right)^{-\gamma_{\beta}} \tag{1}$$

100

115

The subscript "ref" denotes the values of the parameters at the reference height  $z_{ref}$ ,  $\gamma_{\beta}$  is the backscattering hygroscopic growth coefficient. The reference height is usually chosen at altitudes with low RH, where particle can be considered as quasi-dry. However, for deliquescent particles, as mentioned, RH<sub>ref</sub> should exceed corresponding deliquescence point.

The size distribution of atmospheric aerosols is commonly described using a superposition of lognormal distributions, typically corresponding to three distinct modes: nucleation, accumulation, and coarse. Each mode can be characterized by its modal radius,  $r_0$ , standard deviation,  $\sigma$ , and number concentration, N. Increase of the modal radius with RH, assuming that  $\sigma$  is unchanged, can be parameterized as (Randriamiarisoa et al., 2006):

$$\frac{r_0}{r_{0,ref}} = \left(\frac{1 - RH}{1 - RH_{ref}}\right)^{-\varepsilon} \tag{2}$$

Where  $\varepsilon$  is the size growth coefficient, which depends on aerosol chemical composition. The volume of particles in accumulation mode is predominant for urban aerosol and smoke, considered in this study. Thus, the volume of the particle relatively to its value,  $V_{ref}$ , at the reference height is:

$$\frac{V}{V_{ref}} = \left(\frac{r_0}{r_{0,ref}}\right)^3 = \left(\frac{1 - RH}{1 - RH_{ref}}\right)^{-3\varepsilon} \tag{3}$$

The particle volume, modeled with Eq.3, will hereafter be denoted as  $V^{mod}$ . Assuming that total volume, V, is the sum of volume of the dry particle and of water uptake, the real part of the refractive index can be parameterized as (Randriamiarisoa et al., 2006; Raut and Chazette, 2008):

$$m_{R} = m_{R}^{H2O} + (m_{R,ref} - m_{R}^{H2O}) \left( \frac{1 - RH}{1 - RH_{ref}} \right)^{3\varepsilon}$$
 (4)

where  $m_R^{H2O}$  is the real part of the refractive index of water; The real part of the refractive index, modeled using Eq.4, will hereafter be denoted as  $m_R^{mod}$ . When the vertical profile of particle volume density retrieved from  $3\beta+2\alpha$  observations,  $V^{3+2}$ , is available, the ratio  $\frac{V}{V_{ref}}$  in Eq.3 can be replaced by  $\frac{V^{3+2}}{V_{ref}^{3+2}}$  allowing to calculate the  $m_R$  without the knowledge of RH profile:

130 
$$m_R = m_R^{H2O} + (m_{R,ref} - m_R^{H2O}) \frac{V_{ref}^{3+2}}{V_{s+2}^{3+2}}$$
 (5)

The real part of the refractive index, calculated using Eq.5, will be denoted as  $m_R^V$ . The volume  $V^{3+2}$  can only be retrieved for altitudes above the complete geometrical overlap height. Therefore,  $z_{\rm ref}$  must be selected at or above this height. Recall also, that Eqs. (3)-(5) are applicable only within a well-mixed layer. The real part of the RI at reference height, in Eqs. (4) and (5) can be selected from the values retrieved from  $3\beta+2\alpha$  observations, which will be denoted as  $m_R^{3+2}$ . Further in the text, when calculating  $V^{mod}$  with Eq.3 we will use volume  $V_{ref}^{3+2}$  at reference height as  $V_{ref}$ .

If water uptake does not alter the chemical composition and does not cause fluorescence quenching, the fluorescence signal should be proportional to the volume of dry particles. Thus, the variation of the fluorescence backscattering coefficient,  $\beta_F$ , relatively to its value at the reference height,  $\beta_{F,ref}$ , allows for modelling  $m_R$  within the layers, where particle number density varies with height, assuming the particle type remains unchanged. If number density of particles at heights z and  $z_{ref}$  is N and N

140  $N_{ref}$  respectively, their ratio is  $\frac{N}{N_{ref}} = \frac{\beta_F}{\beta_{F,ref}}$ . The real part of the RI calculated per a single particle is:

$$m_{R} = \frac{m_{R,ref} \frac{V_{ref}^{3+2}}{N_{ref}} + m_{R}^{H2O} \left( \frac{V^{3+2}}{N} - \frac{V_{ref}^{3+2}}{N_{ref}} \right)}{\frac{V^{3+2}}{N}} = m_{R}^{H2O} + (m_{R,ref} - m_{R}^{H2O}) \frac{V_{ref}^{3+2}}{V^{3+2}} \frac{\beta_{F}}{\beta_{F,ref}}$$
(6)

The real part of the RI, modeled with Eq.6, will be denoted as  $m_R^F$ .

Our approach to assessing the potential effect of water uptake on fluorescence involves analyzing episodes of strong hygroscopic growth within the well-mixed PBL. A decrease in fluorescence backscattering with height may indicate the fluorescence quenching. Additionally, we compare the profiles of the real part of the refractive index, retrieved from  $3\beta+2\alpha$  observations,  $m_R^{3+2}$ , with the values  $m_R^{mod}$ ,  $m_R^V$  and  $m_R^F$  obtained from Eq.4, 5, and 6 respectively. Agreement between  $m_R^{3+2}$  and  $m_R^{mod}$  would support the applicability of the assumption that the total particle volume results from the additive contributions of dry aerosol and water uptake. Consistency between  $m_R^{mod}$  and  $m_R^V$  would further validate the RH profile used in the calculations. And finally, a significant deviation of  $m_R^F$  from  $m_R^V$  at high RH could suggest the presence of fluorescence quenching.

## 3 Numerical simulation of the particle volume and surface density as a function of the extinction coefficient

We begin with a numerical simulation to explore the relationship between particle volume and surface area density, on one hand, and the extinction coefficient, on the other, for different particle sizes and refractive indices. In this simulation, we consider a set of lognormal particle size distributions (PSDs) for the number density n(r):

$$\frac{dn(r)}{d\ln r} = \frac{N}{(2\pi)^{1/2}\ln\sigma} \exp\left[-\frac{(\ln r - \ln r_0)^2}{2(\ln\sigma)^2}\right]$$
 (7)

where *N* is the total particle number (*N*=1 in all computations) and  $\sigma$  is the standard deviation. The modal radius,  $r_0$ , and  $\ln \sigma$  varied within the [0.015-6.3]  $\mu$ m and the [0.43-0.936] intervals respectively. The particle extinction coefficient at 532 nm,  $\alpha_{532}$ , was computed for spheres with the real and the imaginary part of the refractive index within the ranges [1.3-1.7] and [0.00-0.03] respectively. In total, more than  $10^5$  extinction coefficients were calculated, using the look-up-tables (Kolgotin et al., 2023).

The results of the computations are summarized in Fig.1, which shows particle volume and surface area density as a function of the extinction coefficient at 532 nm. To provide insight into the particle sizes corresponding to different  $\alpha_{532}$ , the upper axis presents approximate values of the effective radius,  $r_{eff}$ . Among the numerous data points shown in Fig. 1, some of them represent the hygroscopic growth of specific aerosol types. To emphasize these points, we overlay lines depicting the hygroscopic growth of SU (sulfate) and OC (organic carbon) particles from the MERRA-2 model (Chin et al., 2002; Colarco et al., 2010). The particle parameters used in the computations, such as modal radius and the RI, are summarized in Table A1 for different relative humidities. The  $ln\sigma$  in Eq.7 was set to 0.79 for OC and 0.71 for SU independent of RH.

The ratio  $\frac{V}{\alpha_{532}}$  in Fig.1a exhibits a tendency to increase with particle size, and for a given value of  $\alpha_{532}$  this ratio varies

significantly, spanning up to an order of magnitude. However, for specific aerosol types, such as SU and OC, the dependence  $V(\alpha_{532})$ , which characterizes particle hygroscopic growth, can be reasonably approximated by a linear fit:  $V=0.38\alpha_{532}$ ,  $V=0.24\alpha_{532}$  respectively, where volume is expressed in  $\mu\text{m}^3\text{cm}^{-3}$  and the extinction coefficient in Mm<sup>-1</sup>. These modeling results suggest that, even in the presence of hygroscopic growth, aerosol-type-specific conversion factors can still enable estimation of particle volume from the extinction coefficient. The variation of  $\frac{S}{\alpha_{532}}$  in Fig.1b is significantly smaller compared to the

 $\frac{V}{\alpha_{532}}$  in Fig.1a, and it decreases with particle size, allowing for a more accurate estimation of surface area density.

Figure 1: Modeled particle (a) volume and (b) surface area density as functions of the extinction coefficient at 532 nm. Cyan and green lines correspond to hygroscopic growth of sulfates (SU) and organic carbon (OC) particles from MERRA-2 model. The particle parameters used in the computations are described in the text.

#### 4 Results of observations

## 4.1 25 March, 2022. Hygroscopic growth within the well-mixed PBL

We begin with the episode on 25 March, 2022, when the fluorescence backscattering remained relatively stable within the PBL. For this episode, measurements were performed at a 48° angle to horizon and the height of complete geometrical overlap of the laser beam and the telescope's field of view is of 700 m. As shown in Fig. 2, variations in WVMR within the 650–1750 m height range remain below 0.3 g kg<sup>-1</sup>, while potential temperature deviations are within 1.5 K. These conditions indicate that the PBL is well mixed. The relative humidity increases with height from 35% to 75%, resulting in a 1.5-fold increase in  $\beta_{532}$ . Using parameterization (1), the backscattering growth coefficient,  $\gamma_{\beta}$ , is estimated to be 0.45±0.1. This result is consistent with observed  $\gamma_{\beta}$  values for different aerosol types, as reported by Sicard et al. (2022). Specifically, for urban aerosols at 532 nm wavelength,  $\gamma_{\beta}$  ranges between 0.38 and 0.65, which aligns well with our findings. During hygroscopic growth, the particle depolarization ratio,  $\delta_{532}$ , decreases with height from 10% to 5%, while the extinction Ångström exponent (*EAE*), calculated from  $\alpha_{355}$  and  $\alpha_{532}$ , decreases from 1.4 to 0.9. The lidar ratios at 355 nm and 532 nm,  $S_{355}$  and  $S_{532}$ , during hygroscopic growth increase from 44±7 sr to 53±8 sr and from 52±8 sr to 69±10 sr respectively. The fluorescence capacity at 700 m is  $G_F$ =0.65×10<sup>-4</sup> and according to the classification scheme in Veselovskii et al. (2024), this corresponds to urban aerosol. This classification is further supported by HYSPLIT Backward Trajectory Analysis, which indicates that the air masses at this altitude originated from Germany.

The profile of particle volume, retrieved from  $3\beta+2\alpha$  observations,  $V^{3+2}$ , is presented in Fig.2c, along with the corresponding conversion factors  $C_{\alpha,532}^V$  and  $C_{\beta,532}^V$  shown in Fig.3a. The conversion factor  $C_{\alpha,532}^V$  remains relatively stable during hygroscopic growth, with a deviation from the mean value of about 10%. As shown in Fig.2c, the volume,  $V^{\alpha}$ , calculated from  $\alpha_{532}$  as  $V^{\alpha} = C_{\alpha,532}^V \times \alpha_{532}^V$ , aligns well with  $V^{3+2}$ , when using a mean conversion factor  $C_{\alpha,532}^V = 0.15 \, \mu \text{m}^3 \text{cm}^{-3} \text{Mm}$ . On the other hand, the conversion factor  $C_{\beta,532}^V = 0.15 \, \mu \text{m}^3 \text{cm}^{-3} \text{Mm}$ . On the index. Fig.2c also displays the profile of  $V^{mod}$ , modeled using Eq.3. The value  $V_{ref}^{3+2}$  is taken from inversion of  $3\beta+2\alpha$  measurements at reference height  $z_{ref}=700 \, \text{m}$ . The size growth coefficient,  $\varepsilon$ , reported by Randriamiarisoa et al. (2006) for urban aerosol in Paris was 0.26. However, for our observations a better fit with  $V^{3+2}$  was achieved when using  $\varepsilon=0.33$ .

Figure 2: Vertical profiles of the particle parameters on 25 March 2022 for period 19:10-21:30 UTC. (a) The backscattering coefficient,  $\beta_{532}$ , the fluorescence backscattering coefficient,  $\beta_F$ , and the Ångström exponent, EAE. (b) The particle depolarization ratio,  $\delta_{532}$ , the water vapor mixing ratio, w, and the relative humidity, RH. (c) The particle volume,  $V^{3+2}$ , retrieved from  $3\beta+2\alpha$  observations; the volume,  $V^{\alpha}$ , calculated from extinction coefficient  $\alpha_{532}$  and the volume  $V^{mod}$  modeled with Eq.3 using  $\varepsilon$ =0.33. (d) The real part of the refractive index,  $m_R^{3+2}$ , retrieved from  $3\beta+2\alpha$  observations, along with the values  $m_R^{mod}$ , modeled with Eq.4 and the values of  $m_R^V$  and  $m_R^F$  calculated from Eqs. (5) and (6) respectively. Measurements were performed at 48 deg to horizon.

The real part of the refractive index,  $m_R^{3+2}$ , retrieved from  $3\beta+2\alpha$  observations, in Fig.2d decreases with height from 1.57 to 1.44 and it agrees with Hänel parametrization,  $m_R^{mod}$ , calculated with Eq.4. The profiles of  $m_R^V$  and  $m_R^F$  calculated with Eqs. (5) and (6) respectively, align with  $m_R^{3+2}$  within the uncertainty of retrieval. Thus, for this specific case, we see no evidence for the fluorescence quenching by the water uptake.

The  $3\beta+2\alpha$  observations enable the estimation of key characteristics of the particle size distribution, at least, for fine-mode particles. As the maximum available laser wavelength is 1064 nm, the retrievals exhibit decreasing sensitivity to particles with radii larger than approximately 2  $\mu$ m (Veselovskii et al., 2009). The evolution of  $\frac{dV}{d \ln r}$  with height is shown in Fig.3b. At 700

m, the fine mode, with a maximum at r=0.15  $\mu$ m, is dominant. By 1700 m the maximum of the PSD shifts to approximately r=0.25  $\mu$ m due to hygroscopic growth. A small secondary maximum at r=1.0  $\mu$ m also appears; however we cannot conclusively determine whether it is a real feature or merely an artifact of the retrieval process. From this figure, we can conclude that, for the episode considered, the hygroscopic growth primarily affects fine-mode particles.

Figure 3: (a) Vertical profiles of extinction-to-volume,  $C_{\alpha,532}^V$ , and backscattering-to-volume,  $C_{\beta,532}^V$ , conversion factors on 25 March, 2022. (b) The particle size distribution  $\frac{dV}{d\ln r}$  at heights 700, 1285 and 1700 m.

## 4.2 30-31 August, 2022. Hygroscopic growth within not well-mixed layer

On 30-31 August, 2022, an episode of pronounced hygroscopic growth was observed. The vertical profiles of the particle parameters, similar to those shown in Fig. 2, are presented in Fig.4 for 30 August, 2022. Complete geometrical overlap occurred at 1250 m, therefore, the extinction profiles and inversion results are available only above this height. The lidar ratios at 1250 m are  $S_{355}$ =57±8 sr and  $S_{532}$ =51±8 sr. The WVMR shows significant height variations within the 1250-2500 m range, indicating that this layer is not well-mixed. The HYSPLIT Backward Trajectory Analysis indicates that the air masses below the 1000 m height originated from the north of Germany, while at higher altitudes they arrived from the Atlantic region and may have transported maritime particles. The relative humidity increases steadily with height and peaking at approximately 85% near 1750 m. Above this level, RH gradually decreases. Within the 1000–2500 m range, aerosol particles undergo hygroscopic growth, and the peak in the backscattering coefficient,  $\beta_{532}$ , is consistent with the RH maximum at 1750 m. In contrast, both the Ångström exponent and the linear depolarization ratio exhibit distinct minima at this height. As previously noted, the aerosol layer is not well-mixed, and the backscattering growth coefficient,  $\gamma_{\beta}$ , can only be derived within specific altitude ranges. For the lower range of 750–1000 m (45%

Figure 4: Similar to Fig.2 but for the measurements on 30 August 2022 during period 19:30-20:30 UTC.

## Figure 5: Similar to Fig.3 but for the measurements on 30 August, 2022 during period 19:30-20:30 UTC.

The retrieved  $m_R^{3+2}$  decreases with height from 1.52 to 1.39 within the 1250-1750 m range, and then begins to increase above this level. The values of  $m_R^V$  and  $m_R^F$  calculated via Eq.5 and Eq.6., as shown in Fig.4d, agree with  $m_R^{3+2}$  within the uncertainty of retrieval. Thus, in this episode, once again, water uptake does not exhibit a significant effect on the fluorescence signal.

The modification of the PSD during hygroscopic growth is shown in Fig.5b. The maximum of the fine mode, within the 1300-1800 m range, shifts with height from *r*= 0.14 μm to *r*= 0.2 μm. Simultaneously, the second mode, centered near 1.0 μm, increases with height. This suggests that the aerosol mixture likely contains coarse hygroscopic particles, which could have a maritime origin.

### 4.3 Possibility of fluorescence quenching by the water uptake

The assumption that fluorescence backscattering is independent of water uptake is crucial when using the fluorescence technique to study aerosol hygroscopic growth (Miri et al., 2024). However, in some episodes we observed a decrease in  $\beta_F$  with height within the well-mixed PBL, which may indicate possible fluorescence quenching. Two such episodes, on 11 May and 25 June, 2024, are shown in Fig.6. The complete geometrical overlap was achieved at 1030 m and 1300 m respectively, with lidar ratios at these heights measured as follows:  $S_{355}$ =50±7 sr,  $S_{532}$ =42±6 sr on 11 May and  $S_{355}$ =46±7 sr,  $S_{532}$ =49±7 sr on 25 June 2024. In both cases, the PBL is well mixed, with variations in WVMR remaining below 0.2 gkg<sup>-1</sup>. The air masses within the 500-2500 m range for both episodes originated from Germany. The fluorescence capacities at 750 m for these cases are  $0.86 \times 10^{-4}$  and  $0.75 \times 10^{-4}$  respectively, which is higher than for previous episodes. The backscattering growth coefficients,  $\gamma_{\beta}$ , on 11 May and 25 June, 2024 are estimated as  $0.4 \pm 0.1$  and  $0.9 \pm 0.2$ . In contrast to results in Fig.2a, the fluorescence backscattering,  $\beta_F$ , decreases with height along with  $\delta_{532}$ . This decrease begins at relatively low RH, and the rapid increase in  $\beta_{532}$  at high RH is not accompanied by a similarly fast decrease in  $\beta_F$ .

Figure 6: Vertical profiles of aerosol and fluorescence backscattering coefficients,  $\beta_{532}$  and  $\beta_F$ , along with water vapor mixing ratio, w, and the particle depolarization ratio,  $\delta_{532}$ , measured on (a) 11 May 2024, (b) 25 June 2024. (c) The relative humidity for the episodes considered.

Figure 7: Similar to Fig.2 but for the measurements on 12 June, 2023 for the period 21:00-22:20.

All episodes demonstrating the fluorescence quenching happened when urban aerosols presented within the PBL. Among these episodes, one of the most representative occurred on 12 June, 2023. Corresponding profiles of particle parameters are shown in Fig.7. Variations in WVMR and potential temperature within the 1000-2750 m range are small (below  $\pm 0.2$  gkg<sup>-1</sup> and  $\pm 2$ K

respectively) indicating that the PBL is well-mixed. The HYSPLIT Backward Trajectory Analysis indicates that the air masses within the 1000-3000 m height range originated from Eastern Europe. The fluorescence capacity at 750 m is quite high  $(1.4\times10^{-4})$ , suggesting the presence of organic particles. The height of complete geometrical overlap is 1500 m and the lidar ratios at this height are  $S_{355}=37\pm6$  sr and  $S_{532}=38\pm6$  sr.

Within the 1000-2700 m altitude range the backscattering coefficient,  $\beta_{532}$ , shows a consistent positive response to RH, reaching its maximum value at 2700 m where RH peaks. The backscattering growth coefficient,  $\gamma_{\beta}$ , is determined to be 0.55±0.1. Simultaneously, the observed increase in RH correlates with reduced fluorescence backscattering, suggesting potential fluorescence quenching effects. The aerosol volume,  $V^{\alpha}$ , derived from  $\alpha_{532}$  measurements shows good agreement with  $V^{3+2}$ , when applying a mean conversion factor  $C_V^{\alpha} = 0.15 \, \mu \text{m}^3 \text{cm}^{-3} \text{Mm}$ . The modeled volume profile,  $V^{mod}$ , calculated using Eq.3 and  $\varepsilon = 0.33$ , similarly matches  $V^{3+2}$  within the layer, where aerosols are presumed to be well-mixed.

The real part of the refractive index,  $m_R^{3+2}$ , retrieved from  $3\beta+2\alpha$  observations, decreases from 1.58 to 1.45, within the 1500-2700 m altitude range. However, this observed decrease occurs at a lower rate compared to both  $m_R^{mod}$  and  $m_R^V$  values, suggesting that the simplified model, which assumes additivity of dry aerosol and water uptake, may not be applicable for this episode. Furthermore, the values,  $m_R^F$ , derived using the fluorescence signal in Eq.6, are systematically lower than both  $m_R^{mod}$  and  $m_R^V$ , providing additional evidence for potential fluorescence quenching effects. A similar behaviour of  $m_R^{3+2}$ ,  $m_R^V$  and  $m_R^F$  was observed for the episodes shown in Fig.6 with corresponding results presented in the Appendix (Figs. A1 and A2).

Fig.8 shows the PSDs for three episodes, discussed in this section. As the particles undergo hygroscopic growth, the fine mode shifts toward larger radii; however, the second mode at r=1  $\mu$ m, observed in Fig.5b, is absent. This suggests that only fine particles are undergoing hygroscopic growth.

Figure 8: The particle size distribution  $\frac{dV}{d \ln r}$  at different heights for three case studies showing evidence of fluorescence quenching during hygroscopic growth: (a) 12 June, 2023; (b) 11 May, 2024; (c) 25 June, 2024.

To quantify fluorescence quenching effects we computed the relative reduction in fluorescence backscattering using:

$$\Delta \beta_F = \frac{\beta_{F,ref} - \beta_F}{\beta_{F,ref}} \tag{8}$$

Fig.9a displays the normalized  $\beta_{532}$  and fluorescence reduction  $\Delta\beta_F$  as a function of RH for three observational periods: 25 March 2022, 12 June 2023 and 11 May 2024. For consistent comparison,  $\beta_{532}$  values were normalized to the backscattering coefficient at RH<sub>ref</sub>=50% and  $\Delta\beta_F$  were calculated for the same RH<sub>ref</sub>. These cases were selected based on two criteria: consistent  $\beta_{532}$ (RH) growth patterns across all events and distinctive variations in  $\Delta\beta_F$  behavior between cases. As previously noted, the 25 March 2022 case showed no significant RH-dependent increase in  $\Delta\beta_F$ . In contrast, during the other two episodes (12 June 2023 and 11 May 2024),  $\Delta\beta_F$  exhibited a steady increase from baseline conditions reaching a maximum reduction of ~50% at RH = 85%. However, the rapid growth of  $\beta_{532}$  at high RH is not mirrored by the same rate of increase in  $\Delta\beta_F$ . This decoupling between hygroscopic backscattering growth and fluorescence suppression suggests that water uptake affects aerosol fluorescence through mechanisms beyond simple dilution effects.

To further investigate water uptake effect on fluorescence, Fig.9b presents the relationship between fluorescence reduction  $\Delta\beta_F$  and particle volume  $V^{\alpha}$ , calculated from  $\alpha_{532}$ , for the same three case studies as in Fig.9a. On 12 June, 2023 and 11 May, 2024  $\Delta\beta_F$  demonstrates sharp increase within ~5–12  $\mu$ m³cm⁻³ volume range and a subsequent plateau at higher volumes. These observations reveal two key findings: quenching starts at relatively low RH (before maximum hygroscopic growth) and the  $\Delta\beta_F$ -volume relationship is nonlinear, disproving simple proportional dependence on water uptake. While aerosol-type dependence is evident, the specific particle composition driving this quenching behaviour cannot yet be definitively identified from these measurements.

Figure 9: (a) The RH-dependence of normalized backscattering coefficient,  $\beta_{532}$  (stars), and fluorescence reduction,  $\Delta\beta_F$  (circles), for three case studies: 25 March 2022, 12 June 2023 and 11 May 2024. All values are normalized relative to RH=50% baseline conditions. (b) The relationship between fluorescence reduction  $\Delta\beta_F$  and particle volume  $V^{\alpha}$ , calculated from  $\alpha_{532}$ , for the same three episodes.

#### 5 Conversion factors for smoke and urban particles

As demonstrated in the previous section, the volume density of urban aerosol, even in the presence of hygroscopic growth, can be estimated using a single extinction or backscattering coefficient, and corresponding conversion factors. However, urban aerosol includes a variety of particle types (e.g. sulfates, soot), causing these conversion factors to vary across episodes. To quantify this variability and assess the resulting uncertainty in V and S calculations, we analyzed multiple episodes of urban aerosol within the PBL exhibiting hygroscopic growth. For each episode, conversion factors at 355 nm and 532 nm were derived from  $3\beta+2\alpha$  observations.

In addition to urban aerosols, we also examined smoke episodes. For smoke, hygroscopic growth events were rare, and our focus shifted to the vertical dependence of conversion factors in the lower and middle troposphere. Fig.10 presents the vertical profiles of the conversion factors  $C_{\alpha,532}^V$  and  $C_{\beta,532}^V$  for ten urban aerosol episodes involving hygroscopic growth, along with six smoke episodes observed between 2020 and 2024.

Figure 10: Height profiles of the extinction-to-volume,  $C_{\alpha,532}^V$ , and backscattering-to-volume,  $C_{\beta,532}^V$ , conversion factors in several measurement sessions for (a, b) urban aerosol and (c, d) smoke.

For urban aerosol, the mean value of  $C_{\alpha,532}^V$  is estimated to be  $0.17\pm0.04~\mu\text{m}^3\text{cm}^{-3}\text{Mm}$ . This contrasts with the higher value of  $0.30\pm0.08~\mu\text{m}^3\text{cm}^{-3}\text{Mm}$  reported by Mamouri and Ansmann (2017) for continental aerosol in Germany. It is interesting that extinction-to-surface area conversion factors  $C_{\alpha,\lambda}^S$ , provided by Mamouri and Ansmann (2016) ( $C_{\alpha,355}^S=1.55\pm0.46$  and  $C_{\alpha,532}^S=2.8\pm0.89~\mu\text{m}^2\text{cm}^{-3}\text{Mm}$ ) well agree with values in Table 1. In contrast to the extinction coefficient, the backscattering

compared to  $C_{\alpha,532}^V$  (Fig.10b). The mean value of  $C_{\beta,532}^V$  is  $10\pm 6~\mu \text{m}^3 \text{cm}^{-3} \text{Mmsr}$ , corresponding to an unacceptably high uncertainty in volume estimation (up to 60%). Consequently, for urban aerosols, reliable volume density retrievals should be based solely on the extinction coefficient.

For smoke, the mean conversion factor  $C_{\alpha,532}^V$  is  $0.13\pm0.02~\mu\text{m}^3\text{cm}^{-3}\text{Mm}$  across all height ranges, which is very close to the value of  $0.13\pm0.08~\mu\text{m}^3\text{cm}^{-3}\text{Mm}$  reported by Ansmann et al. (2021) for aged smoke in AERONET measurements. The composition of smoke is less variable compared to urban aerosol, allowing for reliable volume estimations even based on the backscattering coefficient (Fig.10d). The mean value of  $C_{\beta,532}^V$  for smoke is  $9\pm1.5~\mu\text{m}^3\text{cm}^{-3}\text{Mmsr}$ , indicating comparable uncertainties in volume estimation from both extinction and backscattering coefficients for the analyzed episodes.

Table 1. Conversion factors for calculation of volume and surface area of smoke and urban particles from the extinction and backscattering coefficients at 355 and 532 nm wavelengths.

|       | $C^{\scriptscriptstyle V}_{lpha,355}$ | $C^{\scriptscriptstyle V}_{lpha,532}$ | $C^{\scriptscriptstyle V}_{eta,355}$ | $C^V_{eta,532}$ | $C_{lpha,355}^{S}$                  | $C_{lpha,532}^{S}$ | $C^S_{eta,355}$                        | $C^{\scriptscriptstyle S}_{eta,532}$ |
|-------|---------------------------------------|---------------------------------------|--------------------------------------|-----------------|-------------------------------------|--------------------|----------------------------------------|--------------------------------------|
|       | μm³cm-³Mm                             |                                       | μm³cm-3Mm sr                         |                 | μm <sup>2</sup> cm <sup>-3</sup> Mm |                    | μm <sup>2</sup> cm <sup>-3</sup> Mm sr |                                      |
| Urban | 0.08±0.03                             | 0.17±0.04                             | -                                    | -               | 1.6±0.3                             | 2.9±0.7            | -                                      | -                                    |
| Smoke | 0.085±0.015                           | 0.13±0.025                            | 4.25±1.20                            | 9±2.0           | 1.3±0.25                            | 1.75±0.4           | 60±20                                  | 125±50                               |

Table 1 summarizes the conversion factors for calculating particle volume and surface area density for the episodes presented in Fig.10. The conversion factors  $C_{\beta,\lambda}^V$  and  $C_{\beta,\lambda}^S$  for urban aerosol are excluded, due to their unacceptably high uncertainties.

For aged smoke, we obtain  $C_{\alpha,532}^S$ =1.75±0.4 µm²cm³Mm, which agrees well with the value 1.75±0.25 µm²cm³Mm reported by Ansmann et al. (2021). These findings demonstrate that in numerous cases, both particle volume and surface density of smoke can be retrieved from either a single extinction or backscattering coefficient, which is - particularly valuable when complete  $3\beta+2\alpha$  measurements are unavailable. It should also be noted that the extinction-to-volume conversion factors  $C_{\alpha,532}^V$  for SU (sulfate) and OC (organic carbon) derived from Fig.1a (0.38 and 0.24 µm³ cm³ Mm, respectively) are significantly higher than the corresponding values in Table 1. This discrepancy suggests that using just SU and OC is insufficient for accurately modeling urban aerosol and smoke, as these aerosols consist of complex mixtures of different particle types.

#### Conclusion

We analyzed Mie-Raman-Fluorescence lidar observations during aerosol hygroscopic growth episodes to investigate water uptake effects on fluorescence backscattering,  $\beta_F$ . The well-mixed PBL serves as a convenient environment for such studies, since  $\beta_F$  should remain constant in the absence of water uptake effects. However, during certain episodes we observed a systematic height-dependent decrease in  $\beta_F$ . This observed  $\beta_F$  reduction is difficult to explain solely by changes in aerosol composition and likely indicates fluorescence quenching. Notably, the decrease in  $\beta_F$  begins at relatively low RH (less than 50%), coinciding with a reduction in the depolarization ratio. Interestingly, while  $\beta_{532}$  shows rapid enhancement at high RH levels, this is not accompanied by an accelerated  $\beta_F$  decrease rate.

All fluorescence quenching episodes has been observed within the well-mixed PBLs of urban aerosols. Fluorescence quenching exhibits strong dependence on aerosol composition, as evidenced by its absence in numerous episodes. However, we cannot yet identify the specific particle type responsible for this quenching effect in this study. It should also be mentioned that when fluorescence measurements are conducted using multiple discrete channels, the ratio of fluorescence backscattering coefficients between these channels remains unaffected by water uptake, even though each individual channel is influenced by it (see Fig. 10 in Veselovskii et al., 2025). This implies that the spectral signatures of fluorescence are preserved during hygroscopic growth.

The particle volume and surface area density, retrieved from  $3\beta+2\alpha$  measurements during analysis of hygroscopic growth episodes were used to validate the feasibility of estimating these parameters from a single extinction coefficient. Our analysis shows that for urban aerosols, the uncertainty in V and S determination remains below 25% when using  $\alpha_{532}$ . Smoke particles demonstrate comparable uncertainties in V and S calculation. The conversion factors show clear dependence on aerosol type, which can be effectively identified through fluorescence lidar measurements. These findings highlight the Mie-Raman-Fluorescence lidar as a promising tool for aerosol characterization.

405 **Data availability**. Lidar measurements are available upon request (philippe.goloub@univ-lille.fr).

**Author contributions**. IV processed the data and wrote the paper. QH performed meteorological analysis. TP and GD performed lidar measurements. PG supervised the project and helped with paper preparation. AK performed numerical simulations and MK developed the software for data analysis.

**Competing interests**. The authors declare that they have no conflict of interests.

#### Acknowledgement

This work was supported by the CaPPA project, funded by the French National Research Agency (ANR) through the "Programme d'Investissements d'Avenir" (PIA) under contract ANR-11-LABX-0005-01, the "Hauts-de-France" Regional Council (project ECRIN), and the European Regional Development Fund (FEDER). We gratefully acknowledge the ESA/QA4EO programme for supporting observation activities at LOA. The contribution of Q. Hu was supported by the ANR (project ANR-21-ESRE-0013, OBS4CLIM). The Russian Science Foundation is also acknowledged for its support under project 21-17-00114.

This work has benefited from the support of the ACTRIS-FR research infrastructure as well as from the Center for Aerosol Remote Sensing (CARS) within the ACTRIS-EU research infrastructure. Finally, we acknowledge the French government under the France 2030 programme and the Initiative of Excellence of the University of Lille for funding and supporting the R-CDP-24-003-AREA project.

450

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

## Appendix

545

Table A1. Particle parameters from MERRA-2 model. The modal radius of log-normal PSD along with the real and imaginary parts of the RI at 532 nm for sulfates (SU) and organic carbon (OC) for different values of RH.

| 1RH, % |                            | SU    |          | OC                         |       |         |  |
|--------|----------------------------|-------|----------|----------------------------|-------|---------|--|
|        | <i>r</i> <sub>0</sub> , μm | $m_R$ | $m_I$    | <i>r</i> <sub>0</sub> , μm | $m_R$ | $m_I$   |  |
| 0      | 0.070                      | 1.430 | 1.00E-08 | 0.021                      | 1.530 | 0.00885 |  |
| 10     | 0.075                      | 1.410 | 8.28E-09 | 0.022                      | 1.503 | 0.00765 |  |
| 20     | 0.081                      | 1.396 | 6.99E-09 | 0.023                      | 1.485 | 0.00684 |  |
| 30     | 0.086                      | 1.386 | 6.13E-09 | 0.024                      | 1.466 | 0.00598 |  |
| 40     | 0.091                      | 1.377 | 5.36E-09 | 0.025                      | 1.450 | 0.00525 |  |
| 50     | 0.097                      | 1.370 | 4.75E-09 | 0.026                      | 1.437 | 0.00464 |  |
| 60     | 0.102                      | 1.365 | 4.32E-09 | 0.027                      | 1.425 | 0.00412 |  |
| 65     | 0.105                      | 1.362 | 4.11E-09 | 0.028                      | 1.419 | 0.00385 |  |
| 70     | 0.107                      | 1.360 | 3.93E-09 | 0.028                      | 1.415 | 0.00368 |  |
| 75     | 0.111                      | 1.358 | 3.72E-09 | 0.030                      | 1.407 | 0.00330 |  |
| 80     | 0.114                      | 1.356 | 3.53E-09 | 0.031                      | 1.399 | 0.00296 |  |
| 85     | 0.118                      | 1.354 | 3.37E-09 | 0.032                      | 1.390 | 0.00252 |  |
| 90     | 0.123                      | 1.351 | 3.15E-09 | 0.035                      | 1.378 | 0.00201 |  |
| 95     | 0.133                      | 1.347 | 2.84E-09 | 0.040                      | 1.363 | 0.00133 |  |
| 99     | 0.151                      | 1.343 | 2.47E-09 | 0.054                      | 1.346 | 0.00055 |  |

Figure A1: Vertical profiles of the particle parameters on 11 May 2024. for period 20:30-23:00 UTC. (a) The particle volume,  $V^{3+2}$ , retrieved from  $3\beta+2\alpha$  observations; the volume,  $V^{\alpha}$ , calculated from extinction coefficient  $\alpha_{532}$  and the volume  $V^{mod}$  modeled with Eq.3. (b) The real part of the refractive index,  $m_R^{3+2}$ , retrieved from  $3\beta+2\alpha$  observations, along with the values  $m_R^{mod}$ , modeled with Eq.4 and the values of  $m_R^V$  and  $m_R^F$  calculated from Eqs. (5) and (6) respectively. Values of  $V^{mod}$  and  $m_R^{mod}$  were calculated for  $\epsilon$ =0.33.

Figure A2.: Similar to Fig.A1 but for the measurements on 25 June 2024 for period 22:00-23:00 UTC. Values of  $V^{mod}$  and  $m_R^{mod}$  were calculated for  $\epsilon$ =0.52.