# Peer review of "Impact of water uptake on fluorescence of atmospheric aerosols: Insights from Mie-Raman-Fluorescence lidar measurements"

_EGUsphere, 2025_

## Author Comment (AC1)

Response to RC1

First of all, we would like to thank Referee for very detailed analysis of our manuscript. In the process of revision, we tried to follow his suggestions.

*The authors use a fluorescence lidar to study the hygroscopic growth of urban aerosol at Lille, France. Interestingly, they report periods where the fluorescence backscatter coefficient is not affected by increasing RH and other periods where fluorescence quenching occurs, i.e., a decrease of the fluorescence backscatter with increasing RH. These are important findings because they imply that the fluorescence backscatter coefficient can not always be used to normalize the backscatter enhancement due to hygroscopic growth in not well mixed aerosol layers. Furthermore, the authors report extinction-to-volume and extinction-to-surface-area conversion factors based on the inversion of the lidar data. The applied methods are valid and well presented. The study is relevant and deserves publication in AMT after some minor revisions listed below.*

*Major comments:*

*As it remains unresolved which type of urban aerosol is affected by fluorescence quenching, it is desirable to gather as many information as possible on the episodes with and without fluorescence quenching. Probably, this question can not be resolved in this publication, but providing as many as possible information (e.g., a lidar ratio) on the aerosol situations observed during the 5 case studies will help future scientists to find patterns.*

The values of lidar ratios and overlap heights for all episodes are added to the revised manuscript.

*What about the occurrence of pollen during your observations? The fluorescence quenching cases are from May and June and might be linked to a certain pollen species?*

The main pollen season in Lille occurs between March and April, though pollen episodes can also occur in May. Using fluorescence and depolarization measurements, we took special care to confirm that pollen did not contribute to aerosol backscattering during the studied episodes.

*You provide backscatter hygroscopic growth parameters (gamma_beta) for your case studies (except for 30 Aug 2022, please add). Please compare these values to literature values, which might help to further constrain the aerosol type.*
Hygroscopic growth parameters for different aerosols are summarized in the paper of Sicard et al. We added comparison with this paper.

*Fluorescence quenching is a central topic of your manuscript. Please add some sentences to describe the phenomenon. In the introduction the term is mentioned, but not explained.*
*A usual task for a reviewer is to ask for uncertainty estimates. I would appreciate if you put an uncertainty range to all values mentioned in the text.*

Ln.50. This paragraph is modified as following:
"The addition of a fluorescence channel to Mie-Raman lidar (Veselovskii et al., 2020) opens new opportunities for particle characterization. If hygroscopic growth has no impact on aerosol fluorescence properties, fluorescence may serve as a proxy for evaluating the volume of dry aerosol. However, laboratory studies demonstrate that fluorescence can be suppressed ("quenched") by interactions with other molecules through processes of collision and energy transfer (Lakowicz, 2006), leading to reduction in emission intensity. In particular, water

molecules can efficiently quench fluorescence of organic fluorophores as reported by Dobretsov et al. (2014) and Maillard et al. (2021). Meanwhile, the extent to which such quenching occurs in atmospheric aerosols during hygroscopic growth remains an open question (Gast et al., 2024; Reichardt et al., 2025). Fluorescence quenching is expected to depend on aerosol composition, phase state, and presence of organic coating, highlighting the need to analyse a diverse set of measurement episodes."

*Minor comments*:

*I would prefer the term "surface area" and not just "surface".*

Changed

*The date format is not consistent throughout the manuscript and the figures. I would prefer the order day month year as it is more logical than month day year.*

Changed

*Section 2 is written very concise. It would be helpful to add a sentence describing how the fluorescence backscatter is used to normalize the hygroscopic growth in case the particle number density changes with height. It is described in previous studies, e.g., Miri et al., AMT 2024, but it would be helpful for the general reader to include a short statement around L 123.*

Eq.6 is expanded to explain normalization.

*L 91-94 It is still a pity that you cannot launch radiosondes from Lille. Have you tried to use the (potential) temperature from GDAS instead the radiosonde from such a distance? Especially for the PBL, the variations to a far away radiosonde station might be tangible. Please discuss.*

Yes, unfortunately RH measurements are not collocated. We used temperature profiles from both GDAS and radiosondes. Corresponding comment is added to revised manuscript.

*Fig 1: Please explain SU and OC in the figure caption.*

 Done

*And why are the model results just presented for a certain range of extinction coefficients?*

Results are shown for 0-99% range of RH. We added Appendix with Table, containing OC and SU parameters used in computations.

*L 179 Does the study provide an estimate of the uncertainty for urban aerosol in Paris? It would be helpful to justify the difference from the well-fitting value which you use.*

Unfortunately, this study does not provide uncertainties.

*L 218 The profile of the particle volume V^alpha should align with V^3+2 because the conversion factor to derive V^alpha was estimated based on V^3+2. Is this correct?*

Yes. The idea is to show that mean value of conversion factor can be applied to the whole profile to get volume with high height resolution.

*L 323-324 From your results, I fully agree that reliable retrievals of the volume density should rely on the extinction coefficient. Please note that the method by Mamouri and Ansmann (2017) is based on the aerosol-type specific backscatter coefficient multiplied with the corresponding lidar ratio. In the case of hygroscopic growth as you present, this method will fail because it uses a fixed extinction-to-backscatter ratio disregarding the change of the lidar ratio due to hygroscopic growth. It was recently shown that the lidar ratio of urban aerosol increases with increasing RH.*

Yes. We agree. The method of Mamouri and Ansmann (2017) works well in elevated smoke and dust layers, where hygroscopic growth is normally not observed.

*Section 5: Actually, the conversion factors for the smoke particles are a bit off topic, because the main part of the manuscript deals with hygroscopic growth of urban aerosol. Nevertheless, I find it very interesting and relevant to include the conversion factors for smoke as well. Extinction-to-surface-area conversion factors: Mamouri and Ansmann, ACP 2016, provides conversion factors for the surface area as well. The continental values of 2.8 (Germany, 532 nm) and 1.55 (Germany, 355 nm) agree well with your factors for urban aerosol.*

 We have added corresponding comment and reference to the manuscript.

*Technical corrections*

*Some references are not correct, e.g., L 22 Burton et al., 2012 (according to your reference list) or L328 Fig 11 does not exist, it is Fig 10*
Corrected

*In the introduction and elsewhere, there are some fragments from a former version of the manuscript, e.g., L 29, L 63, L 269 which lead to incomplete sentences*
Corrected.

*Some abbreviations are not explained (but understandable): L 77 RI, L138 sigma*
Corrected

*L 162 Results of Observations*
Corrected

*L 176 On the second instance in this line, the indexing for the conversion parameter is done wrongly (alpha and V are swapped).*
Corrected

*L 247 Here you use a different unit for the WVMR*
Corrected

*L 229 and Fig 4d: the symbol for the real part of the refractive index is not consistent between text and figure.*
Corrected

*L 312: There is a smoke case from 2020 included in Fig 10., so it will be 2020 – 2024.*
Corrected

---

## Author Comment (AC2)

Response to Referee #2

First of all, we would like to thank Referee for very detailed analysis of our manuscript. In the process of revision, we tried to follow his suggestions.

*RC2: 'Comment on egusphere-2025-2107', Anonymous Referee #2, 06 Jun 2025  reply*
*The manuscript from Veselovskii et al., investigates changes in the aerosol fluorescence properties driven by hygroscopic growth phenomena within well-mixed aerosol layers using multiwavelength Raman and Fluorescence lidar measurements at Lille and lidar data inversion methods. During the period of study (2021 – 2024), they found cases where the fluorescence backscatter coefficient decreases with increase of relative humidity within well-mixed PBL, which may indicate potential evidence of fluorescence quenching by water uptake. Additionally, using the available multiwavelength lidar observations (3β+2α) they calculate useful extinction-to-volume and extinction-to-surface conversion factors for urban and smoke aerosols, facilitating the estimation of the particle volume and surface densities for urban and smoke cases when only the aerosol extinction coefficient profile at a single wavelength is available. Overall, the manuscript is well structured and well written and with high scientific significance making the manuscript suitable for publication in AMT journal after minor revisions.*

*My general comments are:*

*The parametrizations in section 2 are not properly defined, except of parametrization (2) in Line 112, which results in a confusion when parametrization (1) and parametrization (3) appear later in the text (Line 167 for (1) and Line 227 for (3)). Please clarify.*

In the process of revision we strongly modified section 2, to make it clear for reader.

*Section 3: The authors mention in lines 139-141 that they calculate the extinction coefficient using look-up-tables for the particle parameters from Kolgotin et al., 2023 without discussing any relation to relative humidity. However, from line 154 they focus on two aerosol types under hygroscopic growth conditions and they mention that the MERRA-2 model has been used to obtain particle parameters under different relative humidities for the calculation of the extinction coefficients. Please clarify and elaborate the discussion for the above points.*

In Section 3, we calculate the extinction coefficient using look-up tables covering a wide range of particle parameters. Among the numerous data points shown in Fig. 1, some represent the hygroscopic growth of specific aerosol types. To emphasize these points, we overlay lines depicting the hygroscopic growth of SU (sulfate) and OC (organic carbon) particles from the MERRA-2 model. This phrase is added to the text.

*Moreover, I miss the link between the simulation results and the observation results, especially for the conversion factors for smoke and urban particles which deviate from the simulations. I think a dedicated discussion about the simulations should be included somewhere in the text.*

In the end of section 5 we have added the fragment:
"It should also be noted that the extinction-to-volume conversion factors  for SU (sulfate) and OC (organic carbon) derived from Fig.1a (0.38 and 0.24 µm³ cm⁻³ Mm, respectively) are significantly higher than the corresponding values in Table 1. This discrepancy suggests that using just SU and OC is insufficient for accurately modeling urban aerosol and smoke, as these aerosols consist of complex mixtures of different particle types.

*The authors mention in line 130 that the deviation of mRF from mR3+2 would indicate presence of fluorescence quenching. However, in their comparisons in section 4 they also use the modeled mRmod and mRV. Do they also expect deviations between the different forms of the modeled refractive indices? I am not so familiar with this and this may apply to others readers as well, so I would suggest the authors to include a brief discussion on what such deviations would imply/mean.*

Referee 1 asks similar question. To clarify it, in the end of section 2 we added paragraph: "Additionally, we compare the profiles of the real part of the refractive index, retrieved from $3\beta+2\alpha$ observations, $m_R^{3+2}$, with the values $m_R^{mod}$, $m_R^V$ and $m_R^F$ obtained from Eq.4, 5, and 6 respectively. Agreement between $m_R^{3+2}$ and $m_R^{mod}$ would support the applicability of the assumption that the total particle volume results from the additive contributions of dry aerosol and water uptake. Consistency between $m_R^{mod}$ and $m_R^V$ would further validate the RH profile used in the calculations. And finally, a significant deviation of $m_R^F$ from $m_R^V$ at high RH could suggest the presence of fluorescence quenching."

*The info about the full geometrical overlap height is provided only for the case on 30-31 August 2022. What is the full overlap height for the other cases? If it is the same (complete geometrical overlap above 1200 m) then the results for heights below this point should be excluded from the discussion and the affected figures.*

Overlap height varied for different episodes. Corresponding information is added to manuscript.

*In similar figures (e.g. 4 and 7) different symbols are used for the same parameters (see V and mR3+2). Please use a common symbol per plotted parameter to all affected figures.*

Corrected

*Line 351 "we confirm that the process does not alter the fluorescence spectral signature.": I didn't understand how and in which section the authors confirm the above statement about fluorescence quenching. Please elaborate to make it more clear for the reader.*

We modified this paragraph as following:
"However, we cannot yet identify the specific particle type responsible for this quenching effect in this study. It should also be mentioned that when fluorescence measurements are conducted using multiple discrete channels, the ratio of fluorescence backscattering coefficients between these channels remains unaffected by water uptake, even though each individual channel is influenced by it (see Fig. 10 in Veselovskii et al., 2025). This implies that the spectral signatures of fluorescence are preserved during hygroscopic growth."

*Specific comments*

*Lines 25-26 "(Kolgotin et al., 2018, 2023, 2024; Chang et al., 2022; Zhou et al., 2024)": I would suggest to include Veselovskii et al., 2002 (mentioned in line 78) that is used in this study,*
Done

*Line 29 "conversion factors, and .": Is something missing here?*
Corrected

*Lines 37-38 "accompanied by a simultaneous decrease ... refractive index": Could the authors add relevant studies to support this statement and point interested readers to more details?*

We have added reference for work of Hänel (1976). All parameterizations originate from that work.

*Lines 44-45 "Multiwavelength lidar … as a function of RH.": Similarly to previous, comment, could the authors add indicative studies for reference?*

Actually, we don't know publication, where all these parameters were derived as a function of RH. It is definitely possible, but was not done yet. In this study we probably for the first time demonstrated, that decrease of the real part with RH can be calculated from 3+2 measurements.

*Line 60-61 "relationship between particle volume … and the extinction coefficient on the other": I think that this sentence is misleading. To my understanding in section 3 the authors investigate the behavior of particle volume and surface density as a function of the particle extinction coefficient for different particle sizes. I would suggest to rephrase the sentence.*

The sentence is modified

*Lines 63-64 "…the conversion factors and  in the presence of the particle hygroscopic growth.": I don't understand this part. Maybe something is missing? Kindly check and revise.*

Sorry, corrected

*Line 72 "particle depolarization ratios": I would suggest to change to particle linear depolarization ratios*

Changed

*Line 77 "and the RI.": With RI probably the authors refer to refractive index. I think RI is not defined earlier in the text, so please add the full name for this abbreviation.*

We added definition of RI

*Line 84 "…refractive index it is ±0.05.": I would suggest to provide the uncertainty in %, as for the volume and surface densities and imaginary part of RI*

We would prefer to stay as it is, because it is commonly used form to present uncertainty of the real part.

*Line 86: I would suggest to clarify here that the conversion factors that are used in this study, have been derived using the  from the 3β+2α dataset and the retrieved V and S values from the inversion algorithm.*

Done

*Line 99 "deliquescent pattern": I would suggest to add a brief discussion about what this deliquescent pattern is or in general what is the deliquescence phase, before moving to the discussion of the backscatter coef. behavior and the parametrizations.*

This paragraph is modified as following
"Dependence of backscattering on RH can be either monotonic or deliquescent, when backscattering is unchanged up to a certain RH value, called the deliquescence point, where a phase transition occurs from solid to liquid (Carrico et al., 2003; Randriamiarisoa et al., 2006).

For relative humidity exceeding the deliquescence value, the dependence of particle parameters, such as β, mR, and r on RH can be modeled on relative to their values at a some reference height, zref."

*Line 105 "The subscript "ref" denotes the values of the parameters at the reference height zref": reference height with respect to what? Are there any specific criteria/conditions on how the reference height is selected? Does it change from case to case? I think a brief discussion should be included somewhere in the manuscript.*

In our study, to model the particle volume and RI we set the reference height at height of complete geometrical overlap. Corresponding comment is added to manuscript.

*Line 112 "Parameterization (2) can also be…": Does this parametrization come from the same studies cited for Eqs 1-3. If not, kindly cite the studies.*

This section is rewritten in revised manuscript.

*Line 117 "...calculated using Eq.4,..": I think it is Eq. 5, is that correct?*

Yes, corrected

*Lines 120-124: I don't fully understand how the authors conclude to the expression (ratio of particle volume and fluorescence backscatter) that substitutes the particle volume ratio in Eq 5. Could the authors extend the discussion on this?*

In revised manuscript we expanded this equation to explain the derivation.

*Line 138 "... and lnσ varied ...": kindly define the σ*

Done

*Line 155 "Table A1": no table A1 in the text.*

Sorry, it is added.

*Line 178 "...Eq. 4 with zref= 690 m.": How Vref is calculated? Do the authors use the V3+2, like they do for mR,ref (line 118)? Please clarify and add it (maybe at section 2?) in the text.*

Explanation is added to the text and to the Section 2 also.

*Line 191 "zref=700m": why the reference height for mR differs from the one selected for V (690 m; line 178) for the same case?*

No, it is the same 700 m. Sorry. Corrected.

*Figure 6: The particle depolarization ratio has similar color with aerosol backscatter coefficient. I would suggest the authors to update the visualization of particle depol. ratio with e.g. dashed line and/or another color so as to be easily distinguishable from backscatter coefficient. Please harmonize all plots with the updated line/color for depolarization.*

Color is changed

*Moreover, for completeness purposes, I would suggest the authors to include the full suite of plotted parameters for the cases on 11 May and 25 June 2024, as already shown in figures 2 and 4. These 'detailed' figures could be added in an appendix if the authors prefer to keep this subsection as is, but they should be discussed in the main text.*

We have added corresponding plots to Appendix

*Line 247 "gcm-3": not the proper unit for WVMR*

Corrected

*Line 256: Do all the fluorescence quenching episodes happen when urban aerosols are present within the layer? Please clarify.*

Yes. We added sentence to the manuscript.

*Line 258 "1000-1750 m range…": Is this range correct? To my understanding, based on figure 7 the PBL is well mixed up to ~2750 m. Please clarify. Moreover, I would suggest a horizontal line to be added in the plot to point to the upper height limit of well mixed layer conditions.*

Sorry, this is mistake. 2750 m. Corrected

*Line 269 -271: Here the authors compare the mRF with the mRmod and mRV to support the potential fluorescence quenching effect, opposite to what is mentioned in section 2 (~lines 130-131) and discussed in lines 193-194 and 229-230 for the no fluorescence quenching cases. Could the authors explain why do they compare different parameters and extend the discussion in the text?*

Yes, we agree that it should be discussed. We added paragraph to Section 2: ""Additionally, we compare the profiles of the real part of the refractive index, retrieved from $3\beta+2\alpha$ observations, $m_R^{3+2}$, with the values $m_R^{mod}$, $m_R^{V}$ and $m_R^{F}$ obtained from Eq.4, 5, and 6 respectively. Agreement between $m_R^{3+2}$ and $m_R^{mod}$ would support the applicability of the assumption that the total particle volume results from the additive contributions of dry aerosol and water uptake. Consistency between $m_R^{mod}$ and $m_R^{V}$ would further validate the RH profile used in the calculations. And finally, a significant deviation of $m_R^{F}$ from $m_R^{V}$ at high RH could suggest the presence of fluorescence quenching."

*Line 349: To my understanding the fluorescence quenching has been observed within well-mixed layers (PBL) of urban aerosols, but no specific aerosol type has been identified to trigger the quenching effect. I think it should be highlighted the.*

We modified conclusion, to highlight this statement.

*Technical comments*

*Line 50 "for evaluation the volume…": change to "for evaluating the volume…"*
Done

*Line 269 - 270 "Furthermore, the values … signal in Eq. 6, systematically": kindly check for duplication*
Corrected